# Air-conditioning and the adaptation cooling deficit in emerging economies

Filippo Pavanello [1,2], Enrica De Cian [2,3✉], Marinella Davide [2,4], Malcolm Mistry [2,3,5], Talita Cruz [6], Paula Bezerra [6], Dattakiran Jagu [2], Sebastian Renner [7], Roberto Schaeffer [6] & André F. P. Lucena [6]

Increasing temperatures will make space cooling a necessity for maintain comfort and protecting human health, and rising income levels will allow more people to purchase and run air conditioners. Here we show that, in Brazil, India, Indonesia, and Mexico income and humidity-adjusted temperature are common determinants for adopting air-conditioning, but their relative contribution varies in relation to household characteristics. Adoption rates are higher among households living in higher quality dwellings in urban areas, and among those with higher levels of education. Air-conditioning is unevenly distributed across income levels, making evident the existence of a disparity in access to cooling devices. Although the adoption of air-conditioning could increase between twofold and sixteen-fold by 2040, from 64 to 100 million families with access to electricity will not be able to adequately satisfy their demand for thermal comfort. The need to sustain electricity expenditure in response to higher temperatures can also create unequal opportunities to adapt.

[1] University of Bologna, Ca' Foscari University of Venice, Department of Economics, Bologna, Italy. [2] Fondazione CMCC, RFF-CMCC EIEE, Venice, Italy. [3] Ca' Foscari University of Venice, Department of Economics, Venice, Italy. [4] Harvard University, Ca' Foscari University of Venice, Department of Economics, Venice, Italy. [5] The London School of Hygiene & Tropical Medicine (LSHTM), Department of Public Health, Environments and Society, London, UK. [6] Centre for Energy and Environmental Economics, Energy Planning Program, Graduate School of Engineering, Universidade Federal do Rio de Janeiro (CENERGIA/PPE/COPPE/UFRJ), Rio de Janeiro, Brazil. [7] Mercator Research Institute on Global Commons and Climate Change, Berlin, German Institute for Global and Area Studies (GIGA), Hamburg, Germany. ✉email: enrica.decian@unive.it

As global temperatures rise, a growing number of people around the world will be exposed to the potential harm caused by heat stress[1]. Adaptation through the use of air-conditioning[2] has been the subject of a recent and growing literature that looks at patterns of potential needs and demand across major cities[3,4], countries[5,6] and world regions[7,8]. Low- and middle-income countries in the tropics or sub-tropics are under the spotlight[9]. About two to four billion people living in those places have no space-cooling devices in their homes and air-conditioning usage is highly concentrated among high-income households[10]. In a warming climate, air-conditioning could contribute to maintain labor productivity[11–13] and to enhance the accumulation of human capital[14] in the long-term. Better understanding how many of those people at risk will or will not be able to adopt air-conditioning remains an important area for future research.

The adoption of air conditioners follows the "S"-shaped pattern that characterizes the uptake of other durable goods, such as automobiles and refrigerators[15,16]. In developing countries, the growth of this curve tends to start off slowly, because of credit constraints, followed by a steeper rise once income levels reach a certain threshold. Stylized "S"-shaped functions have also been used to project future air-conditioning adoption and energy requirements in India[5] and in other low-income countries[17,18]. The expansion of households' air-conditioning will put increasing pressure on future energy demand especially in hot developing countries[19–21], and accounting for this additional driver of energy demand will help improve the aggregate projections and scenarios needed for managing long-term investments[22–24]. Demand-side actions will be an important element in the transition towards net zero emissions over next few decades[25], but most models used to support policy making lack the characterization of adaptation-energy feedback mechanisms. How energy use for adaptation might influence the design of effective mitigation actions remains to be studied[26,27].

Here we provide a multi-country, comparative analysis of how income and climate drive air-conditioning adoption in Brazil, India, Indonesia, and Mexico, in relation to a comprehensive set of country-specific household characteristics, and evaluate with a top-down approach[28] how future changes in climate and socio-economic conditions centered around 2040 will influence air-conditioning adoption and electricity. We show that in emerging economies the decision to purchase air-conditioning in response to warmer climatic conditions is strongly anchored to a household's socio-economic conditions and demographic characteristics. Not explicitly accounting for other characteristics of households can significantly bias the estimates of the marginal contribution of income and climate, which would appear larger. Although the penetration of air-conditioning is expected to increase in the future, an adaptation cooling deficit, characterized by millions of less well-off electrified households that need but cannot obtain air conditioners, will remain. Increasing the use of electricity for residential space cooling is a form of adaptation that helps relieve the population from heat stress, but the recurring electricity expenditure required limits the opportunities among the lowest income deciles. In the long run, if left to uncoordinated and autonomous actions, space cooling runs the risk of exacerbating local and global negative externalities and of widening existing inequalities.

## Results

**An up-to-date database of households and climate.** Our results are based on the analysis of a new database that combines the up-to-date household-level survey data covering 2172 subnational regions in Brazil, Mexico, India, and Indonesia over the 2003–2018 period, with gridded cooling degree days (CDDs).

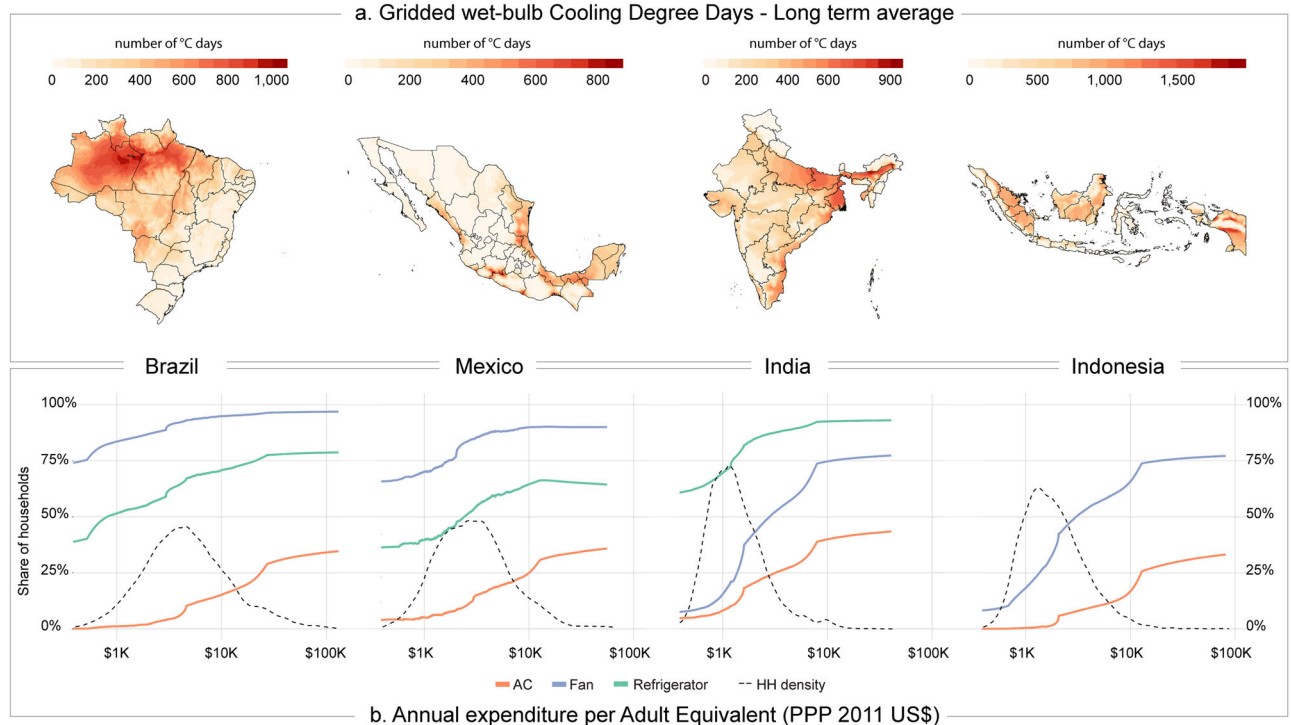

**Fig. 1 Climate, air-conditioning, and income characteristics in four selected emerging economies. a** A 30-year average of gridded wet-bulb cooling degree days (CDDs), up to the second wave of household data used in the study (2009 for Brazil and 2012 for all other countries). **b** Rates of air-conditioning (AC) ownership in relation to per capita total expenditure (2011 US constant dollars at PPP) and comparison to other cooling devices in the second wave of household data. The black dashed line shows the distribution of households (HH) across income levels. Maps are generated using the sp, rgdal, and raster R packages.

We respond to recent demands to account for the influence of relative humidity[7,8] by using wet-bulb temperature as a more accurate measurement of thermal discomfort that, contrary to dry-bulb temperature, does not overestimate temperature at low humidity levels[29]. To better reflect tropical conditions, we use a higher baseline temperature of 24 °C as opposed to the 18 °C value used in most studies on air temperature impacts and building energy demand[30]. Because temperature set-points can vary across households[4], we also consider a lower temperature threshold of 22 °C as a robustness test. The combination of two temperature thresholds with calculations based on dry-bulb and wet-bulb temperatures makes it possible to evaluate the sensitivity of the results for different countries to the climate metric used. For the sake of clarity, in the remainder of this paper, CDDs refer to those computed with wet-bulb temperature and at a base temperature of 24 °C (see section "Climate Data" in Supplementary Information, where results based on CDDs computed with dry-bulb temperature are also shown). Brazil, Mexico, India, and Indonesia are all tropical countries characterized by relatively high average wet-bulb CDDs, though there is significant variation from one country to another (Supplementary Fig. 1). Climate variation remains significant even within each of the four countries considered. The highest long-term average values of wet-bulb temperature are observed in Indonesia and India, although climate heterogeneity between and within countries highlights the presence of high-CDD regions even in Brazil (Fig. 1a). The diffusion of air-conditioning units across districts and states closely mirrors patterns of hot climate conditions in the climate maps, though urbanization and access to electricity play a mediating role (see Supplementary Fig. 1). In India, for example, the highest CDD values, observed in the states of West Bengal, Assam, Uttar Pradesh, and Orissa, are not associated with the most widespread use of air-conditioning. Households in those regions are mostly rural and often lack access to electricity, as implied by low ownership rates of refrigerators. Fans, which consume less energy and do not require a stable connection, are more widespread throughout the country. In Brazil, the state of Rio de Janeiro shows relatively high adoption rates for air conditioners, despite the lower number of annual CDDs compared to its northern states, where urbanization is low. Although Indonesia has the highest values of CDDs, households rarely own air-conditioning units, except for the districts of Jakarta and the Riau Islands.

Climate is only part of the story, as shown by India and Indonesia. For the same level of total expenditure per capita, air-conditioning ownership rates are the highest in India and the lowest in Indonesia (Fig. 1b). In these Asian regions, average annual total expenditure per capita, which we use as an indicator of lifetime income, is below 10,000 USD for nearly all households. The expenditure distribution has a larger variance in Brazil and Mexico where, on average, of at least a quarter of households reports annual total expenditure per capita above 10,000 USD. Across all countries air-conditioning ownership is quite low (12% in India in 2012, 14% in Mexico in 2016), even in Indonesia and Brazil where more recent data are available (8% in Indonesia in 2017, 20% in Brazil 2018). By comparison, fans and refrigerators are more widely used. In India, as early as 2012, fans were owned by 73% of households, even among those with very low-income levels. Refrigerators have the highest adoption rates in Brazil and Mexico (See Supplementary Table 4 for descriptive statistics). Electricity expenditure reflects the ownership patterns of energy-consuming durables. Absolute values are the highest in Brazil and Mexico though, in relative terms, Indian households spend the largest share of their budget on electricity, between 3.4 and 4.5%.

**Drivers of air-conditioning adoption.** We estimate adoption models for air conditioners for each individual country by using the two most recent survey waves available with a logit model (see "Methods"). To understand how adoption patterns differ from more commonly owned goods, we also look at the adoption of refrigerators and fans. While fans can substitute air conditioners in the space cooling service they provide, air conditioners are more comparable to refrigerators in terms of the budget required to purchase them. By using two waves, we can control for country-specific, time-varying unobservable trends that affect all households, such as changes in the prices of appliances and country-level regulations.

Income conditions and climate are both important drivers of the decision to adopt air conditioners across all countries (Table 1), but their relative contribution varies in relation to other household characteristics (Supplementary Table 7). The marginal effect of total expenditure is always larger than that of CDDs (except for fans in Mexico), but climate remains an important factor, especially in Brazil and Mexico. Fans, which in the short-term have the lowest costs, are generally more sensitive to CDDs as compared to air-conditioning. Especially in the warmer countries, India and Indonesia, education and the quality of dwellings correlate with a household's wealth and are more strongly related to the adoption of refrigerators and air-conditioning, the most expensive goods. The extent to which climate affects the decision to adopt also depends on a household's average income level. The interaction term between CDDs and total expenditure (Supplementary Table 5) indicates that households respond to rising temperature levels by purchasing a new air-conditioning unit only when their average annual income is sufficiently high (Fig. 2a). Moreover, as income increases, households tend to substitute fans with air-conditioning. Refrigerators provide a different service that is desirable across all climates but, as income increases, refrigerators become less sensitive to climate. The adoption of refrigerators responds to CDDs at low-income levels in Brazil and Mexico—where adoption is higher—and at medium income levels in India and Indonesia—where adoption is still quite low.

Demographic and infrastructural characteristics are also important factors in explaining adoption patterns, and their relative contribution, compared to income and climate, varies across countries and the type of good considered (Supplementary Table 6). Urbanization increases the probability of adopting cooling durables, and so does home ownership, though this factor is of less importance in comparison to living in major urban centers. Since for Brazil we lack information on districts, regressions only consider households located in the strata of capital and urban regions because for these strata, the geographical climate information are more accurate. The regressions for Brazil, therefore, do not include the urbanization variable. Education substantially enhances the propensity to adopt all types of goods considered in all countries. The housing index, which combines information on the quality of roofs, toilet and walls, shows a positive relationship with adoption propensity, indicating that households occupying higher-quality homes are more likely to install an air-conditioning unit. Demographic factors show a robust influence across goods and countries. Household size has a negative sign, whereas the presence of members under 16 years of age has a positive influence. Households with older family heads are more inclined to have a cooling appliance, probably because such persons spend more time at home. Employed household heads, who spend less time at home, are less interested in owning air conditioners. Findings on gender are mixed, and whether having a male head increases or not the propensity to adopt and use of cooling devices varies across countries. Not including this rich set of households'

**Table 1 Total marginal effects for CDDs wet-bulbs and total expenditure from standardized logit models based on the two most recent waves for air-conditioning (AC), fans (FAN), and refrigerators (REF).**

| | Brazil | | | Mexico | | | India | | | Indonesia | |
|---|---|---|---|---|---|---|---|---|---|---|---|
| | Ac | Fan | Ref | Ac | Fan | Ref | Ac | Fan | Ref | Ac | Ref |
| CDDs | 0.0565*** | 0.0880*** | −0.0029*** | 0.023*** | 0.244*** | 0.014*** | 0.017*** | 0.063*** | 0.031*** | 0.0037*** | 0.073*** |
| | (0.00154) | (0.00359) | (0.00056) | (0.00406) | (0.0126) | (0.00292) | (0.00588) | (0.00783) | (0.00678) | (0.000425) | (0.00750) |
| Tot Exp. (log) | 0.0928*** | 0.0888*** | 0.0167*** | 0.0319*** | 0.119*** | 0.0610*** | 0.0495*** | 0.0930*** | 0.247*** | 0.0123*** | 0.307*** |
| | (0.00148) | (0.00216) | (0.00053) | (0.00276) | (0.00596) | (0.00251) | (0.00259) | (0.00303) | (0.00517) | (0.000465) | (0.00332) |
| Obs. | 75,290 | 75,290 | 75,290 | 78,607 | 78,607 | 78,607 | 167,648 | 170,470 | 166,402 | 524,112 | 524,112 |

Clustered standard errors at district level for MEX, IDN, and IND and region- and year-fixed effect for BRA. State- and year-fixed effects for MEX, IDN, and IND and region- and robust standard errors for Brazil in parentheses. Notes: Interpretation (Brazil). For a representative household, a 1 SD increases in CDDs raises the probability of adopting AC by 5.65 percentage points on a probability scale 0–100. 1 SD increases in the log of income raises the probability of adopting AC by 9.28 percentage points. The total marginal effects include the contribution of the interaction between CDDs and total expenditure and is computed at the mean value of those variables. Full regression results with the full list of covariates are shown in Supplementary Table 7.
***p < 0.001; **p < 0.05; *p < 0.1.

characteristics would significantly bias income and CDD elasticities, which would be estimated to be larger (Supplementary Table 11). Over time, the ability of households to adapt to climate conditions increases. When adoption behaviors are estimated by using only the most recent wave, income and CDD elasticities are significantly larger (Supplementary Table 11), indicating that, for the same income level, climate conditions, as well as all other covariates (*ceteris paribus*), households have a higher probability to adopt air-conditioning in the most recent waves. The higher adaptive capacity of households could also reflect the rapid decline in air-conditioning prices observed over the last twenty years[31], though we cannot formally test this hypothesis with our current data.

While new technologies widen the space of adaptation options available to households, contributing to enhancing their adaptive capacity, actual adaptation depends on behaviors and specifically on how electricity is used. Although we do not observe the specific consumption of electricity for space cooling, we know the total electricity consumption of households. Not only can air-conditioning be reasonably assumed to be more sensitive to changes in temperature than other final usages, but it is also much more energy-intensive compared to fans[32]. Most of the factors that positively influence the adoption of air-conditioning adoption—CDDs, income, urbanization, education, home ownership and housing index—are also positively related to electricity consumption (Supplementary Table 9).

As CDDs increase above historical levels, air-conditioning generally rises more rapidly than fans and refrigerators, especially in Brazil (Fig. 2b). In India and Indonesia, the speed of diffusion aligns with that of other devices. In Mexico, fans reach a saturation point very rapidly, reflecting the relatively higher correlation with CDDs in a country characterized by very heterogeneous climate conditions.

Even within tropical regions, temperature measurements based on dry-bulb temperature can over-estimate CDD elasticities, depending on how air-conditioning is distributed across sub-regions with different micro-climates and humidity levels (Supplementary Table 12). If climatic conditions are measured with dry-bulb CDDs, the estimated CDD elasticities are significantly larger in Mexico and India and only slightly so in Brazil. Mexico and India have a high concentration of air-conditioning in the regions characterized by a particularly arid climate (warm arid and very hot dry climate conditions). Overall, our results are robust in relation to the use of different temperature thresholds, as well as to different measurements.

**Future adoption of air-conditioning around mid-century**. We simulate how changes in future climate and socio-economic conditions will influence a household's air-conditioning adoption and electricity use around 2040 (see "Methods") by combining the change in CDDs simulated under two scenarios of moderate and vigorous warming, as described by the mean climate model representative concentration pathways (RCPs) 4.5[33] and 8.5[34] with changes in income described by five different shared socio-economic pathways[35,36]. In India CDDs increase by a factor of 1.9–2.3, while total expenditure increases by a factor of 4–7 across SSPs. In Indonesia (Brazil), CDDs increase by a factor of 5–9 (6–8) across RCPs while total expenditure by a factor of 3–4 (1.6–2.5) across SSPs. In Mexico CDDs and total expenditure increase by a factor of 1.7–2.5 across SSPs and RCPs.

Increase in the adoption of air-conditioning is substantial (Fig. 3 and Supplementary Tables 18–21). In India, the average adoption rate across Indian states increases from 12% in 2012 to 49–69%, across SSPs and RCPs, in 2040; in Indonesia, from 8% in 2017 to 43–61%, in Mexico from 14% in 2016 to 35–42%, and in

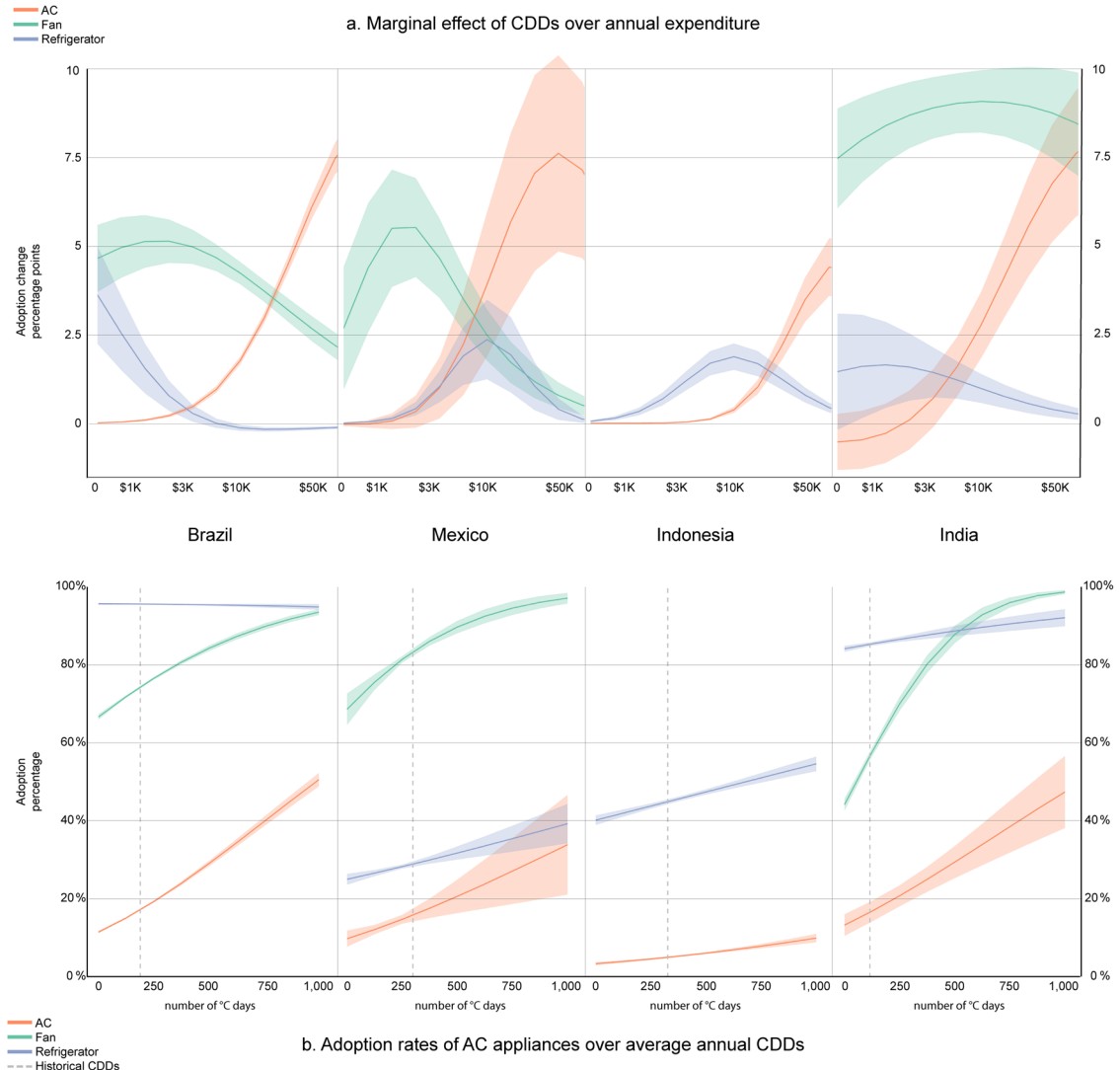

**Fig. 2 Drivers of air-conditioning adoption. a** Marginal elasticity of air-conditioning adoption to a one-hundred increase in CDDs across income levels. **b** Predicted adoption rates of AC and other cooling devices for varying CDDs wet-bulbs. All other drivers are assumed at their historical mean value (full regression results shown in Supplementary Table 5). The vertical dashed line marks the country-specific, long-term historical average of CDDs. Shaded areas represent the dispersion in predicted adoption levels across households.

Brazil from 20% in 2018 to 65–85%. In Brazil, the largest increases are observed in its more affluent states in the southern and southeastern parts of the country, such as São Paulo, where air-conditioning rises from 16 to 78% in SSP5, RCP8.5, and Mato Grosso do Sul, which, starting from 28%, achieves full saturation (90% in SSP5, RCP8.5; results across SSPs and RCPs are available in the Supplementary Material). Brazil's northern states have higher historical ownership rates and therefore see a relatively smaller increase, though they achieve the largest shares by 2040. To mention a few examples, Amazonas, with the contribution of the city of Manaus, Pará, and Tocantins range from 69%, 23%, and 29% in 2018, respectively, to full ownership. In Mexico, the average ownership rates in its hotter states are comparatively high already in the historical records, reaching 73% in Sonora or 77% in Sinaloa. The country's average increase in air-conditioning ownership is mediated by the inland regions, which are characterized by very low CDDs and hence no use of air-conditioning. In India, hetero-geneous conditions in the access to electricity contribute to determining a more diverse situation across states. We do not model expansion in electricity access and therefore our projections represent households that already have access to electricity at

present. This is not an issue for Mexico and Brazil, as they practically coincide with the total survey population (more than 97%). It might lead to an underestimation of AC expansion in Indonesia and India where many households still lack access. The largest increases in air-conditioning are seen in the northeastern part of the country, close to the border with Bangladesh, in states such as Assam, Bihar, Nagaland, and Meghalaya, where CDDs reach the highest values in the country. In India, 6 out of its 35 states, Delhi, Chandigarh, Haryana, Punjab, Rajasthan, and Uttar Pradesh, are expected to achieve full ownership, though only Delhi, Haryana, and Punjab do so across all scenarios. Indonesia exhibits the smallest variation in air-conditioning ownership rates across states. Compared to the other three countries, nearly all states show high CDDs. Still, air-conditioning ownership rates remain relatively low when economic growth is considered. Only Jakarta will come close to full ownership across all scenarios considered in 2040, starting from its 2017 average adoption rates of 30%. Increasing electricity demand also appears to be a ubiquitous form of adaptation (Supplementary Fig. 7), and the interquartile range of the estimated growth factor is always positive (Supplementary Tables 14–17).

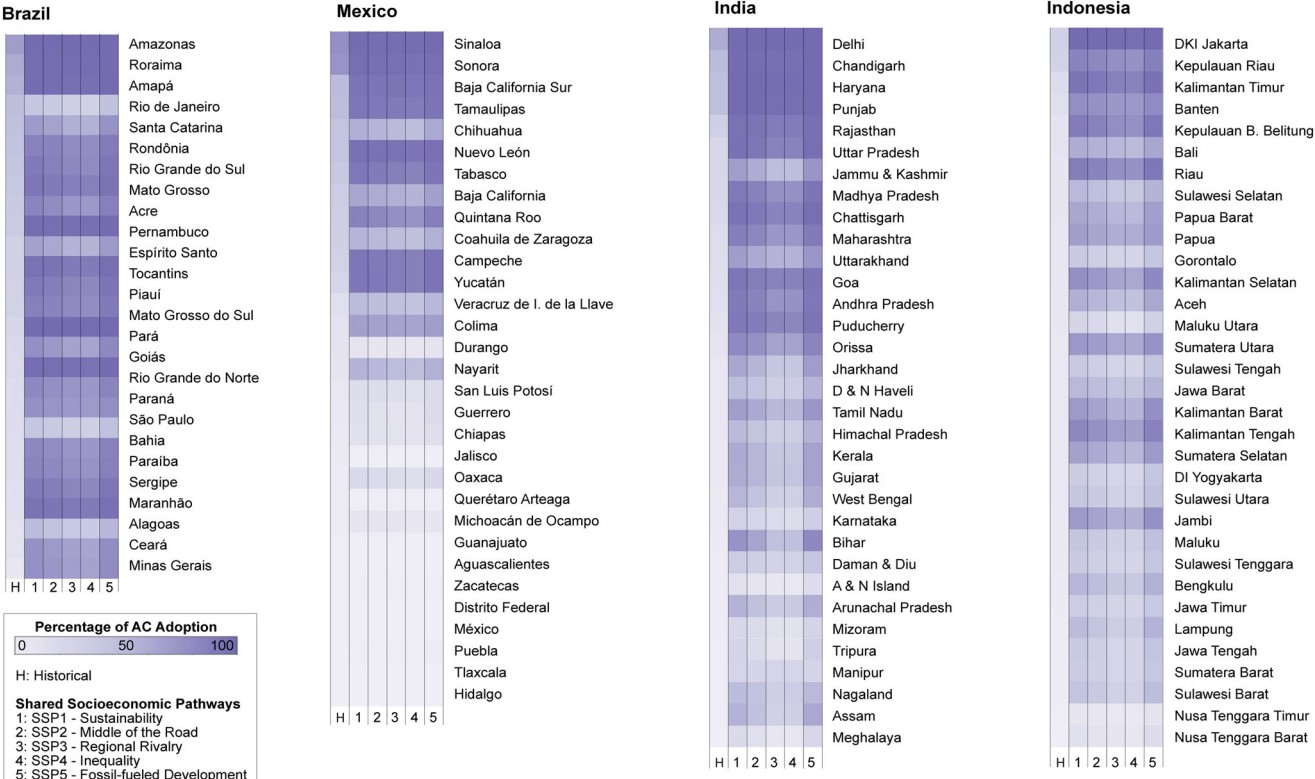

**Fig. 3 Future average air-conditioning adoption rates across country states in 2040 under RCP8.5-warming.** States are ranked from top to bottom, based on historical ownership rates. State-level adoption rates are computed as weighted average of household-level projected adoption rates (see "Methods").

How temperature is measured and how the comfort setpoint is defined are two important sources of uncertainty that could generate different projections, arising from the interaction between the estimated elasticities and the changes in the temperature variables and the associated degree days. When the estimated elasticities are combined with future CDDs, future projected air-conditioning can be lower when using wet-bulb CDDs (Mexico) because of the lower estimated elasticities, but they can also be higher (Brazil) because only slightly smaller elasticities interact with a larger increase in wet-bulb CDDs relative to the historical period compared to dry-bulb CDDs. Since historical wet-bulb CDDs are much lower than dry-bulb CDDs, their growth rate is higher. Projections based on the 22 °C temperature threshold tend to underestimate projections based on the 24 °C temperature, especially when using wet-bulb measurements (Supplementary Table 12 and Fig. 8).

**Adaptation cooling deficit**. Changes in climate and income conditions will allow more households to have an air conditioning unit by 2040, even when considering the uncertainty characterizing future socio-economic conditions. Yet, a non-negligible fraction of the population will be left behind. Our findings show that in 2040, between 64 and 100 million households (in SSP5-RCP8.5 and SSP3-RCP45, respectively) out of the total number of households living in the four countries considered in the latest waves of 343 million will face an adaptation cooling deficit. These households will face climate conditions warmer than their own country average, measured in terms of a country-specific CDD exposure ratio, and yet they will not be able to protect themselves with air-conditioning, as indicated by an air-conditioning availability ratio. We measure total CDD exposure as in Biardeau et al.[7] by multiplying country- and state-level

CDDs by the total number of households. We then compute the CDD exposure ratio for each subnational state across the four countries. When state-level CDD exposure is higher than the country median, the ratio takes a value larger than one and proportional to the distance from the median. This exposure ratio is compared to the AC ratio, which is defined in a similar way. When the state-level average AC ownership rate is smaller than the country median, the ratio takes a value smaller than one, proportional to the distance from the median. When the state-level average air-conditioning ownership rate is larger than the country respective median, the ratio takes a value greater than one and proportional to the distance from the median.

By combining these two ratios, Fig. 4 divides the four countries' states into four groups, for the historical (left panel) and future period (right panel). The imaginary diagonal running from the top-left to the bottom-right quadrant sheds light on the cooling inequality characterizing these countries. States in the top-left quadrant have high adoption rates relative to the country median, despite having lower-than-average CDDs. The state of Rio de Janeiro in Brazil is an example. States in the bottom-right quadrant raise concerns because they have lower-than-average adoption rates despite the higher-than-average exposure to hot climate conditions.

Since socio-economic conditions improve at a faster rate than the increase in CDDs, in comparison with the historical data, the number of states with households experiencing a cooling deficit declines. Brazil and India potentially experience the largest reduction in the adaptation cooling deficit, going from 23 million in 2018 to 8–13 million across the 2040 socio-economic and warming scenarios in Brazil, and from 54 million in 2012 to 29–58 million households in India. In Indonesia, the change is from 26 million households in 2017 to 20–28 million. In Mexico, the historical situation would not change significantly, and it

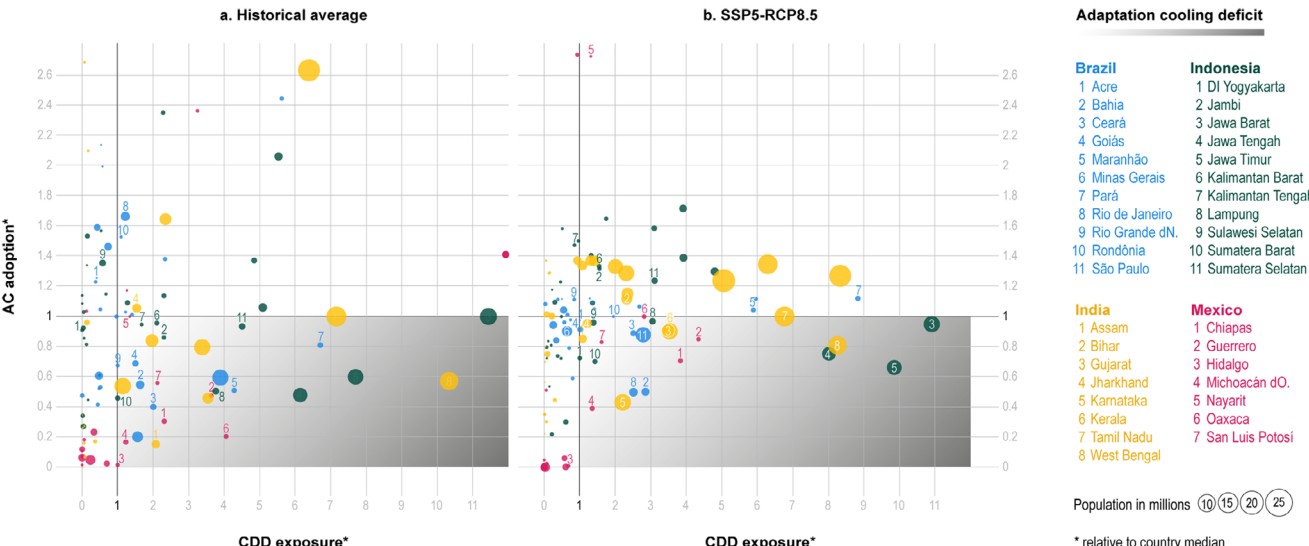

**Fig. 4 Adaptation cooling deficit.** Current situation (**a**, latest wave available) and future projections in 2040 with RCP8.5 warming and SSP5 (**b**) computed with Cooling Degree Days (CDDs). Bubble size proportional to the current number of households relative to each country's maximum. For the historical period, the following waves are used: Brazil, 2018 India, 2012, Indonesia, 2017, Mexico, 2016. Colors are used to differentiate the four countries. See http://www.energy-a.eu/cooling-deficit/ for the interactive online version.

could even worsen (from 5 million in 2016 to 4–6 million households). States with high urbanization levels, hot and humid climate, or with generally poor economic conditions are more likely to face a cooling deficit. Consider, for example, the state of Jharkhand in northeastern India. Because air-conditioning does not keep pace with population and CDDs growth, its position shifts from the top to the bottom-right panel.

The greatest increase in the adoption of air conditioners will be among middle-class and wealthy families, though actual electricity use will rise especially among the wealthiest households (Fig. 5). Electricity use increases with income (Supplementary Tables 9 and 10), though families sharing similar socio-economic conditions might still have very different usage patterns due to building characteristics, appliance efficiency, climate, and infrastructure conditions, which we can only imperfectly account for. The adaptation cooling deficit persists, especially within the lowest income groups. In 2040, median adoption rates in the first total expenditure decile vary between about 1% (SSP3, RCP4.5) and 27% (SSP5, RCP8.5) in India, between less than 0.1 and 40% in Brazil, between 0 and 3% in Mexico, and between less than 0.1 and 5% in Indonesia. The wealthiest households drive the aggregate implications in terms of energy use, which are substantial. Electricity increases by about two to three times in Indonesia and India, while the increase is less dramatic in the Latin American countries (Supplementary Tables 14–17). Results show a higher sensitivity to socio-economic scenarios. The distribution of projected air-conditioning and electricity growth rates are not statistically different across climate scenarios, whereas they are across SSPs.

## Discussion
While rising temperature and increasing income are likely to exert a positive pressure on the adoption and use of air-conditioning, here we show that the dynamics of air-conditioning are country-specific and relate to demographic and infrastructural characteristics, including education and housing conditions. Access to air-conditioning is highly uneven, indicating that households' ability to adapt to climate change through the use of energy is linked to their socio-economic conditions.

The empirical evidence obtained for Brazil, India, Indonesia, and Mexico contrasts in three key respects with the result from the more studied wealthier countries. First, income has a comparatively more important role than climate in explaining the adoption of air-conditioning, and income critically determines a household's ability to respond to increased exposure to CDDs. By contrast, findings from more developed countries suggest that climatic conditions play a relatively larger role in comparison to income[37–39] since, on average, industrialized countries are above the income threshold at which CDD elasticities rise. Second, better educated heads of households have a consistently stronger propensity to adopt and use air-conditioning. This finding may suggest that the influence of better education goes hand in hand with income and is not associated with a greater awareness of the environmental implications of using air-conditioning, which is in contrast with what is found in richer countries. Third, the relative role of urbanization is an important factor in air-conditioning use, though it plays a smaller role in Brazil, India, Indonesia, and Mexico than in the OECD countries.

With respect to the role of relative humidity, we show that projections based on CDDs computed with the 24 °C wet-bulb temperature threshold lead to higher adoption rates and increases in electricity demand compared to simulations based on lower temperature thresholds or dry-bulb-temperature (Supplementary Fig. 8). India and Mexico are two exceptions. We conclude that whether temperature measurements based on dry-bulb-temperature lead to larger or smaller elasticities and projections depends on how air-conditioning is distributed across sub-regions with different micro-climates and humidity levels, and therefore is country-specific. Moreover, the higher density of wet-bulb CDD distribution around small values, especially in Brazil and Mexico, contributes to determining a wider dispersion in the simulated rates of future adoption and electricity consumption.

Aggregate results are in line with the evidence provided by recent single-country studies, such as Gertler and Davis[5] for Mexico. We extend to India, Indonesia, and Brazil concerns regarding a potentially enormous impact from air-conditioning. Over the next twenty years, demand for air-conditioning could rise rapidly with income and CDDs, if households will adjust as they have been doing in the recent past, and so will their demand

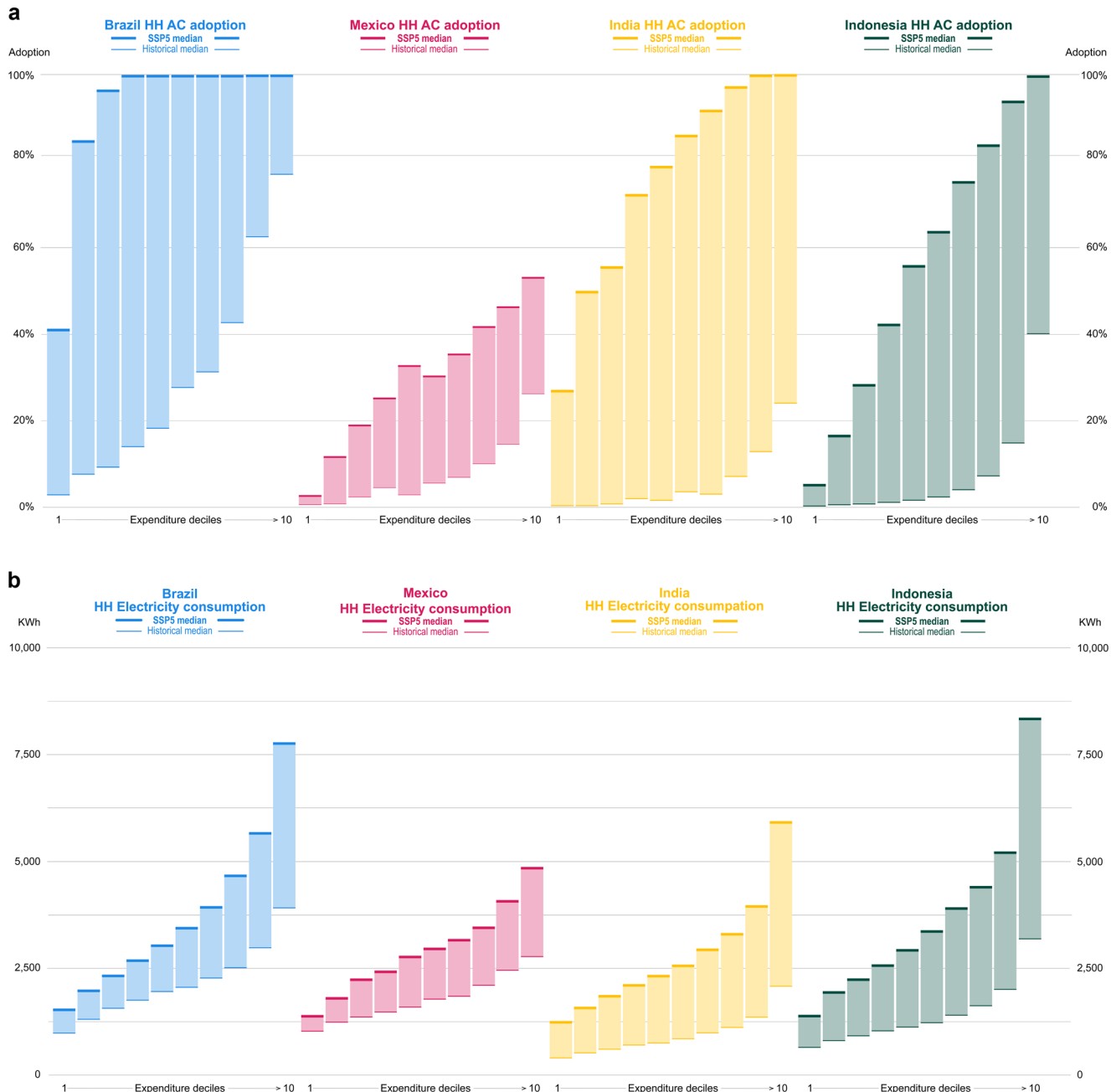

**Fig. 5 Future increase in air-conditioning and electricity use.** Air-conditioning adoption rates (**a**) and total final electricity use (**b**) by income decile in the SSP5 RCP8.5 scenario (historical values refer to the latest available wave, Brazil, 2018; India, 2012; Indonesia, 2017; Mexico, 2016). Horizontal lines show the historical (thin line) and future (thick line) median share across states, as influenced by changes in total expenditure and CDDs. Colors are used to differentiate the four countries and shaded areas highlight the increase between today and 2040.

for electricity. Average electricity growth factors vary across SSPs and RCPs: between 1.3 and 1.8 in Brazil, 2.4 and 3.5 in India, 2.3 and 3.2 in Indonesia, and 1.4 and 1.9 in Mexico, with most of the variation driven by differences across socio-economic scenarios (SSPs), and not so much by differences across climate scenarios (RCPs). Urbanization, education, housing conditions, and electrification, which are taken as given in the simulations, can only further amplify these trends, unless structural changes modify their relationship with mechanical space cooling.

We emphasize that these countries have a vast unmet demand for air-conditioning, and that the uneven distribution of economic resources prevents less affluent households from acceding to this means of adaptation. In 2040, these four countries taken

together will face a cooling deficit of up to almost 100 million households, considering only those that already have access to electricity. Not only will the cooling deficit persist for a non-negligible fraction of the population, but even those with air-conditioning will be exposed to a new condition of vulnerability related to supply shortage in the power sector[40] or degraded power stability[41]. It is therefore imperative to manage the growing appetite for residential space cooling by using a mix of technology-oriented and behavioral or social measures and policies[42,43].

Multiple sources of uncertainties will play out over the next twenty years, layered on top of climate and socio-economic uncertainties, and we account for them by utilizing combinations

of models and scenarios (see "Methods"). Behavioral adaptive responses themselves can change, as suggested by the way our estimated elasticities vary not only across countries, but also over time. These differences can reflect changes in technology, characteristics of infrastructure, and market conditions, all of which contribute to propagating uncertainty. Although our database makes it possible to check for a wide set of a household's characteristics, unobserved elements, such as culture, institutions, can always bias cross-sectional estimates. Electricity costs, as well as appliance costs, certainly play a role in a household's decisions concerning adoption and utilization. Our estimates can only include fixed effects that are meant to capture the influence of a state's fixed characteristics, as well as time varying factors common to all states within each country. Higher elasticities obtained when estimating results only with the latest waves could indeed suggest that unobserved declining costs of appliances have made adoption easier over time.

Our simulations for the future focus on the potential influence of CDDs and income without considering the further adjustments that could be induced by the evolution of prices, technology, and by structural changes. Our estimates for air-conditioning adoption and electricity demand can be used as inputs by quantitative system models to analyze the macroeconomic consequences induced by the simultaneous adjustments across multiple sectors. Integrated assessment models (IAMs) or computable general equilibrium models (CGEs) can also be used to examine the tension between adaptation and mitigation in terms of economic costs, welfare implications, policy effectiveness, and design.

## Methods

**Empirical analysis**. McFadden's basic utility framework (1974, 1982)[44,45] provides the theoretical framework describing the adoption behavior of households. The utility of household $i$ is modelled as a function of expenditure and ownership of goods under the budget constraint given by the household's resources. We distinguish between a vector of cooling durables, $k_i$ with price $p$ and expenditure on all other items $c_i$:

$$U_i = U(c_i, k_i)$$
$$\text{s.t. } c_i + p'k_i = y_i \qquad (1)$$

A household's preferences with respect to the decision to purchase a cooling durable goods are revealed by the latent variable $k_{i,j}^*$ with $\in \{AC, FANS, REF\}$, which can be modelled as a function of a vector of explanatory variables $Xi\beta$ and a random independent error term, $\varepsilon_j$:

$$k_{ij}^* = X_i\beta + \varepsilon_j \qquad (2)$$

The latent variable is revealed once adoption of a given technology is observed. We model the decision to adopt a cooling durable as a dichotomous variable, $k_j$, determined by the following decision rule:

$$k_{ij} = \begin{cases} 1 \text{ if } k_{ij}^* > 0 \\ 0 \text{ otherwise} \end{cases} \qquad (3)$$

and the probability of a household's purchasing device $j$ as a logistic function:

$$P(k_{ij} = 1 | X) = \frac{\exp(Xi\beta)}{1 + \exp(Xi\beta)} = \Lambda(Xi\beta) \qquad (4)$$

where $\Lambda()$ is the logistic cumulative distribution function.

In our specification we want to focus on the relative contribution of climate and income, proxied by total expenditure and their interaction. Following a number of studies evaluating the electricity-temperature response function in Brazil[46] and India[47], as well as that of AC ownership[5] showing how adjustments in electricity demand to climate change vary with income, we assume that the marginal effect of CDDs on the adoption of cooling assets depends on the level of income ($y$). The marginal effect of income, approximated by total household expenditure, also depends on climatic conditions:

$$P(k_{ij} = 1 | CDD, y, Xi) = \Lambda(\beta_1 CDD + \beta_2 y_i + \beta_3 CDD y_i + X_i\beta) \qquad (5)$$

$$\frac{\partial P(k_{ij} = 1 | CDD, y, Xi)}{\partial CDD} = \Lambda(.)'[\beta_1 + \beta_4 y_i] \qquad (6)$$

$$\frac{\partial P(k_{ij} = 1 | CDD, y, Xi)}{\partial y_i} = \Lambda(.)'[\beta_2 + \beta_4 CDD] \qquad (7)$$

This specification implies that the marginal effects of climate and income are not constant.

The CDD-response function of electricity consumption is estimated for each individual country by applying Ordinary Least Squares (OLS) with a sandwich cluster estimator to the most recent wave available for each country. We model electricity use in average annual kilowatt-hours for each household, $q_i$, as a function of CDDs, income, $y_i$, and a set of control variables, $X_i$:

$$\ln(q_i) = \beta_1 CDD_i + \beta_2 y_i + \beta_3 CDD_i y_i + X_i\beta + \epsilon_i \qquad (8)$$

By omitting the ownership of air-conditioning and other energy-using appliances, the model captures the long-term response of electricity use to climate and income, as discussed in Depaula and Mendelsohn[46]. Not including air-conditioning, fans, and other appliances means that we are assuming they can change over time, and the effect of the changes in these variables is implicitly captured by the coefficient of the CDD variable. The energy demand literature has long made a distinction between the so-called intensive margin, i.e. how electricity demand varies with temperature for a given stock of equipment, and the extensive margin, namely how the adoption of appliances changes with temperature, income, and other covariates. Earlier studies discuss how the two decisions are jointly related, and how not accounting for common determinants can lead to biased estimates[48]. Unfortunately, the data gathered for the four countries do not make it possible to develop a two-stage approach that accounts for the short-term effect of air-conditioning on electricity consumption, as in Randazzo et al.[38]. We can, therefore, only evaluate the long-term responses.

**Data**. We build a household-level database using survey data over the 2003–2018 period for four emerging and developing countries—Brazil, India, Indonesia, and Mexico. Three waves are available for Brazil, Indonesia, and Mexico, including the most recent years (2016–2018), whereas only two waves are available for India. In Table 1, we estimate adoption models for air conditioners, fans, and refrigerators for each individual country by using the two most recent survey waves available. The use of two waves makes it possible to include time dummies that check for country-specific, time-varying unobservable variables. However, our projections, as shown in Figs. 3–5 for both air-conditioning and electricity, are based on regression results that only use the most recent wave, since it better reflects the most recent conditions of these fast-growing countries. Supplementary Table 11 shows the sensitivity of CDDs and total expenditure elasticities when different waves are used.

**Validation**. We evaluate the predictive power of our logit models by using the area under the receiver operating characteristic curve (AUC and ROC)[49]. The most important component of our model is the AC adoption model, which is based on a logistic regression that studies determinants of a dichotomic outcome, such as having or not having air-conditioning. Validation techniques for approaches based on logistic regressions exploit a classifier algorithm. Predicted probabilities are computed for all observations, and then the classifier algorithm assigns each predicted probability to class 0 or 1, based on a threshold (usually 0.5). If the predicted probability is larger than 0.5 the observation is classified in class 1, namely as having air-conditioning. If the predicted probability is smaller than 0.5 the observation is classified in class 0, namely as not having air-conditioning. The results are predicted classes for all the observations that are subsequently compared with the truly observed classed, in order to check the accuracy of the model. The justness of a logistic regression is evaluated by building a confusion matrix, a table of fitted versus observed observation classes that makes it possible to identify, after choosing the classification threshold, the number of false positives and negatives that the model predicts. Since the threshold choice for classification is arbitrary, the validation practice computes such a confusion matrix for multiple thresholds and visualizes the results by using a ROC curve displaying the two types of errors for all possible thresholds. The overall performance of the logistic regression is evaluated over an infinite number of thresholds by computing the area under the ROC curve, called AUC. The AUC has a value of between 0.5 and 1. The larger the AUC the better the performance of the logistic regression. We first train our logistic regression on a training dataset defined as a random subsample of our dataset—containing 3/5 of total observations—and then we predict households with air-conditioning in the test dataset, as the remaining subsample of 2/5 of total observations. For three countries, the ROC exhibits an area under the curve (AUC) of more than 0.9 (it is 0.83 which is still very good in Brazil) for air-conditioning, and more than 0.8 for both fans and refrigerators (Supplementary Fig. 2). This suggests a good performance of our models in predicting owners of a cooling asset.

**Projections**. We use the shared socioeconomic pathways (SSPs) and representative concentration pathways (RCPs), a set of five socioeconomic and GHG emission scenarios that have been developed by the research community to make scenario-based mitigation and impact studies more comparable across the literature[36]. The socio-economic scenarios (SSPs) describe five plausible and internally consistent storylines, named SSP1 to SSP5, that narrate how socio-economic variables might unfold over the century[35]. Representative concentration pathways (RCPs) are

trajectories of total future radiative forcing that have been used as input by climate models that generate projections of temperature and other climate variables[36].

We use nationwide growth rates of per capita gross domestic product (GDP), considering a long-term average GDP per capita between 2020 and 2060, and assuming that household expenditure will increase at the same rate. In all countries, per capita GDP grows the most in SSP1 and SSP5, followed by SSP2, which is the continuation of historical trends. Growth rates are particularly high for India (between 289 and 528% compared to 2010) and Indonesia (263–409%), whereas in Mexico and Brazil GDP per capita approximately doubles.

Projections of future dry-bulb ($CDD_{db}$) and wet-bulb ($CDD_{wb}$) cooling degree-days are obtained by using two different sources of meteorological variables from climate model simulations. Data for bias-corrected daily mean temperature dry-bulb ($T_{db}$) for 2021–2060 mid-century are from NEX-GDDP. NEX-GDDP is a broad combination of downscaled and biased-corrected 0.25 gridded daily meteorological fields from 21 global climate models (GCMs) that simulate vigorous (RCP 8.5) and moderate (RCP 4.5) warming under the coupled model intercomparison, phase V (CMIP5) climate modelling exercise. Because the NEX-GDDP does not include projections of humidity, $CDD_{wb}$ are computed by using variables from the ISIMIP2b scenarios[50], which include bias-corrected data[51] from four CMIP5-models over the same period and for the same two RCP scenarios (GFDL-ESM2M, HadGEM2-ES, IPSL-CM5A-LR, MIROC5). The multi-model median $CDD_{wb}$ of the four GCMs (21 GCMs in the case of $CDD_{db}$) are then utilized for the subsequent aggregation to sub-national levels in each country. The projections of the subnational-level population weighted degree-days for the four countries use population data from Jones and O'Neill[52], who provide decadal population (2020–2100) at 0.125° gridded resolution, for the five SSPs. We utilize projected populations for the year 2040 in each SSP as being representative of the midpoint of our mid-century projections. The 0.125° gridded population is matched to the gridded degree-days by using CDO remapping operators, with a prior weighting of the degree-days by population to the district level boundaries in R[53].

To predict the percentage of households with air-conditioning we estimate a logit model by using the latest available wave for each country. We then replace each household's current total expenditure and CDDs with the projected CDDs and expenditure around the year 2040. Projected CDDs are computed by applying state-level growth rates to the historical (as simulated by climate models 1986–2005) district-level CDDs. Projected household-level expenditure is computed by scaling up household expenditure with the country-level income growth projected by different SSPs. We use the fitted equation from the logit model to calculate the adoption probability for each household. In Fig. 4, to estimate the future number of households with air-conditioning, we used the 0.5 probability cutoff. Figure 5 shows state-level averages in air-conditioning ownership rates by expenditure decile computed from the household-level adoption rates. To predict future household-level electricity demand, we have fitted the estimated OLS equations with updated income and CDD values, keeping all other covariates to their historical value. The increase in electricity demand shown in Fig. 5 and in Supplementary Tables 9–10 has been computed at the household level, and then aggregated to the state level by taking the mean value.

## Data availability

The output data generated in this study are available in the Github repository: [https://github.com/Energy-a/Comparative_paper_NatComms]. No access code is required and the following DOI can be used for citation: [https://zenodo.org/badge/latestdoi/363125121]. This repository also contains R-scripts to regenerate all figures in this paper. An interactive visualization of the adaptation cooling deficit is available at [http://www.energy-a.eu/cooling-deficit/]. The input data used in this analysis are available at in the Data Mendeley repository: [https://data.mendeley.com/datasets/ws7cmwbnfg/1] and can be cited using the following https://doi.org/10.17632/ws7cmwbnfg.1. Additional raw input data used in this analysis are available at the following public locations: NASA/NOAA GLDAS: [https://disc.gsfc.nasa.gov/datasets/GLDAS_NOAH025_3H_2.0/summary?keywords=GLDAS_NOAH025_3H_2.0]; CMIP5-NASA NEX GDDP climate data: [https://www.nccs.nasa.gov/services/data-collections/land-based-products/nex-gddp]; ISMIP: [https://esg.pik-potsdam.de/projects/isimip2b/]; GDP and population for the Shared Socioeconomic Pathways: [https://tntcat.iiasa.ac.at/SspDb]. Spatial population data for the historical period: [https://beta.sedac.ciesin.columbia.edu/data/set/gpw-v4-population-count-rev10]; Spatial population projections for the SSPs: [https://doi.org/10.7927/H4RF5S0P]. The raw data for Indonesia are protected and are not available due to data privacy laws.

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

## Acknowledgements

This research was supported by the ENERGYA project, funded by the European Research Council (ERC), under the European Union's Horizon 2020 research and innovation program, through grant agreement No. 756194. Roberto Schaeffer and André F.P. Lucena would also like to acknowledge the financial support received from the National Council for Scientific and Technological Development (CNPq), and from the National Institute of Science and Technology for Climate Change Phase 2, through CNPq Grant 465501/2014-1, and the National Coordination for High Level Education and Training (CAPES), through Grant 88887.136402/2017-00, all from Brazil. The authors are also grateful to Teresa Randazzo for her suggestions, to Karen Bardon for editing the paper, and to Jacopo Crimi for editing the figures. Any remaining errors are those of the authors.

## Author contributions

F.P. and E.D.C designed and performed the analysis. F.P. and M.D. gathered, processed, and harmonized household data and wrote the database description in the Supplementary information, and with the contribution of P.B., T.C., D.J., S.R., A.L., and R.S. M.M. processed climate data. E.D.C. wrote the first draft of the paper. All contributed to editing the paper.

## Competing interests

The authors declare no competing interests.
