## [Peer Review File · Nature Communications]

Air-Conditioning and the Adaptation Cooling Deficit in Emerging EconomiesREVIEWER COMMENTS

Reviewer #1 (Remarks to the Author):

Undoubtedly the topic is interesting and has significant importance considering the present context. My expertise lies in energy-climate coupling and details of the energy systems.

1) Usually Energy, Climate and Social coupling are evaluated using Integrated Assessment Models (IAM). These models comprehensively capture the complex coupling between Energy, Climate and Human systems by using simplified models for future climate, energy and human system models [1]. Although these models are often used to consider generation pathways, there are instances that they focus on the demand side as well [2]. I don't find any connectivity of the present study with such an IAM and it is difficult for me to understand how far accurate the models are when representing this complex coupling.

2) The second major concern is validation. Although validation of such integrated assessment procedures that link energy, future climate and cities is difficult [3] it is important that the authors provide sufficient confidence to the readers that the results obtained do make sense. For example, some of the assumptions made by the authors relating to cooling and energy demand for cooling are often used at top down models but lead to significant errors when you compute them using bottom up models. More importantly, such HVAC (heating ventilation and air-conditioning) demands are strongly influenced by the building physics which has not been considered at all. There are IAMs that use top down approaches. However, a clear validation about the model (if not all the elements but at least certain important parts within the model) is essential.

3) Quantification of uncertainties

Climate and energy related solutions always come up with uncertainties. In this case, future predictions for climate variation, technology evolution, market conditions can come up with many uncertainties where uncertainty propagation plays a major role [4]. However, the model does not discuss these uncertainties comprehensively.

Considering these limitations, I cannot recommend the publication to be published in Nature Communication as it is.

References

[1] Moss RH, Edmonds JA, Hibbard KA, Manning MR, Rose SK, Vuuren DP van, et al. The next generation of scenarios for climate change research and assessment. *Nature* 2010;463:747–56. <https://doi.org/10.1038/nature08823>.

[2] Grubler A, Wilson C, Bento N, Boza-Kiss B, Krey V, McCollum DL, et al. A low energy demand scenario for meeting the 1.5 °C target and sustainable development goals without negative emission technologies. *Nature Energy* 2018;3:515–27. <https://doi.org/10.1038/s41560-018-0172-6>.

[3] IAM helpful or not? *Nature Climate Change* 2015;5:81–81. <https://doi.org/10.1038/nclimate2526>.

[4] Perera ATD, Nik VM, Chen D, Scartezzini J-L, Hong T. Quantifying the impacts of climate change and extreme climate events on energy systems. *Nature Energy* 2020;5:150–9. <https://doi.org/10.1038/s41560-020-0558-0>.

Reviewer #2 (Remarks to the Author):

This study presents an empirical investigation on households adoption of air-conditioning in relation to climate and socioeconomics in four developing countries, and future scenarios. The paper advances the current state of the art by providing important insights on distributional aspects and inter- and intra-country differences in air-conditioning adoption and future space cooling deficit in developing countries. The manuscript is of interest for a broader audience and policy-relevant, clear and well written. My main comments are about comparison with a similar study, some methodological concerns, and strengthening of the conclusions.

Major comments:

- The authors should acknowledge that a similar empirical study exists for one of the investigated countries, Mexico (see reference below). It is recommended to revise the introduction and conclusions to highlight the advancements compared to this study and, if possible, add results comparison.

Reference: Lucas W. Davis, Paul J. Gertler. Air conditioning and global warming. *Proceedings of the National Academy of Sciences* May 2015, 112 (19) 5962-5967; DOI: 10.1073/pnas.1423558112

- The choice of using higher baseline temperatures for CDD calculation and use of wet-bulb versus dry-bulb CDDs are reasonable, in agreement with the indications in existing literature. However, I could not find in the text any justifications for the choice of 24°C as baseline temperature for both dry-bulb and wet-bulb CDDs. Using different baselines might lead to different results, depending on how temperature and humidity levels trigger air-conditioning adoption. I would therefore suggest to test how different baseline temperatures affect the model results and check to which extent results are robust across

temperature levels and different countries. This could provide important additional insights and support the baseline temperature selection.

- Results based on both dry-bulb and wet-bulb CDD are provided, however there is no discussion on differences in outcomes between the two and whether accounting for humidity provides improved estimates in AC adoption in different countries.

- L.77. "We show that the growing penetration of air-conditioning will undoubtedly cause an upward pressure on aggregate electricity use". This claim seems not entirely supported by the results, since the electricity model does not explicitly include air-conditioning adoption and it is not clear to what extent the increase in electricity demand might be driven by air-conditioning versus other appliances adoption.

- L.278. "Growth factors vary significantly, from around threefold (across SSPs and RCPs) in Brazil...". However, from Figure A.8 it seems that SSPs have a much greater influence on electricity changes than RCPs. The authors are invited to further elaborate on this (also related to my previous point) and report results for RCP4.5 in addition to RCP8.4 in Tables A.13-A.16.

- A paragraph comprehensively describing the main limitations of this study and future research directions is missing in the discussion/conclusions section, and its addition is suggested.

Minor comments:

- L. 101: "households rarely use air-conditioning units". Would "own" be more appropriate?

- L.124: "We estimate adoption models for air conditioners for each individual country by using the two most recent 124 survey waves available with a logit model". Please, justify and provide more details in the methods section on how multiple survey data were used in the estimation of the air-conditioning adoption model.

- Table 1: please, define in the caption the acronyms AC, FAN, and REF.

- Fig.5. It is suggested to specify in the caption that electricity use is total (not only air-conditioning).

Reviewer #3 (Remarks to the Author):

Overall, I found the new dataset collected and analyzed here to be a novel contribution, but a better elucidation of some of the strategies of the analysis and conclusions drawn would constitute an improvement in the paper.

The major claim of the paper is that it 'brings new evidence on the specificity of air-conditioning dynamics and electricity use in Brazil, India, Indonesia and Mexico'. It also focuses on using this analysis to project future uptake of air conditioning, as well as the 'adaptation cooling deficit' presented by persistent low incomes and exacerbated by climate change. The collection and combination of extensive survey data for each country studied seems to be a new and important contribution to the field. The conclusions state that 'both climate and income have comparatively similar roles in the increased adoption of air-conditioning in three of these countries. By contrast, findings from more developed countries suggest climate conditions play a relatively larger role than income...as well as the role of other variables.' This type of conclusion strikes me as not particularly novel and indicates a more general aspect that it is not totally clear what important insights are made available by the large dataset and sophisticated statistical analysis presented by the paper.

For instance, Table 1 is a highly simplified distillation of adoption dynamics into only two variables (climate and income), and appears to tell most of the story. What's less clearly presented is the relative importance and value of the many other variables included in the regressions, many of which (e.g. education level, urbanization, access to electricity) are clearly highly correlated. More discussion of this, and possibly Annex might help clarify this. Given the inherent uncertainties in forecasting economic development and climate, in my opinion the goal of such an analysis is to minimize the number of variables and clearly explain the explanatory value of those chosen and their limitations.

A more minor point, I was somewhat confused by the use of both fans and refrigerators included for comparison as 'cooling durables', since the service provided by these two are very different. Refrigerators are intuitively highly desirable in all climates, while fans are a space cooling 'proxy' for air conditioners and thus should have some degree of anti-correlation with them. Table 1 seems to indicate that the dependence on refrigerators on CDD is statistically significant, which is counterintuitive, and needs explanation.

In the forecasting, a potentially significant variable in the forecasting of cooling deficit would be the real (PPP) cost of AC equipment, which has decreased rapidly in the past 20 years. This could be presented essentially as a sensitivity analysis.

Finally, the placement of Materials and Methods at the end of the paper is somewhat strange to me. It might be clearer to put an abbreviated version of this up front, and include abbreviated details of dataset, functional forms and statistical results in the appendix.

We thank the reviewers for their feedback and suggestions that have helped to improve the earlier version of the manuscript. We address the points raised by the reviewers below. In addition, we have also provided all data and code used in the study for replication as follows:

- (i) Input data files:
- (ii) STATA and R codes for replication + output data files (including figures and tables):
https://github.com/Energy-a/Comparative_paper_NatComms and
<https://data.mendeley.com/datasets/ws7cmwbnfg/1> (data)

In addition, we also plan to make the final key results available through an online interactive mode that will be published on this page (now password protected) <http://www.energy-a.eu/cooling-deficit/>

Reviewer #1

Undoubtedly the topic is interesting and has significant importance considering the present context. My expertise lies in energy-climate coupling and details of the energy systems.

1) Usually Energy, Climate and Social coupling are evaluated using Integrated Assessment Models (IAM). These models comprehensively capture the complex coupling between Energy, Climate and Human systems by using simplified models for future climate, energy and human system models [1]. Although these models are often used to consider generation pathways, there are instances that they focus on the demand side as well [2]. I don't find any connectivity of the present study with such an IAM and it is difficult for me to understand how far accurate the models are when representing this complex coupling.

REPLY: We thank the reviewer for this comment, which gave us the opportunity to better explain how our study connects with IAMs.

At the onset, it is worth clarifying that ours' is not an IAM study. The underlying framework proposed by this study 1) has potential values as it stands and 2) delivers results that can be used to parameterize demand-side climate change adaptation in IAMs or CGEs or even bottom-up models, which is still a major gap in state-of-the-art IAMs.

The bulk of the existing and new empirical literature on climate change impacts and climate adaptation on energy demand looks at developed countries, mostly the US and Europe (see Auffhammer & Mansur (2014) for a review). A few global studies exist, but the ability to statistically identify significant impacts in tropical countries has been limited by the aggregate nature of the data. The empirical literature is bringing more and more evidence that the marginal effect of climate conditions does depend on the pre-existing climate conditions. This evidence questions approaches that extrapolate the response functions from some countries, most notably the US, to other countries for which there was a lack of empirical evidence. An example on this point is precisely the estimation of an air-conditioning penetration model using data from US cities, though adjusted for country specific income (McNeil & Letschert, 2010; Sailor & Pavlova, 2003) that has been used in global models (Isaac & van Vuuren, 2009), as well as in single-country studies for other countries (Akpınar-ferrand & Singh, 2010).

Our ambition here is to estimate two models, one for AC adoption and one for electricity demand specific for a set of emerging countries that could be used to parameterize AC adoption and climate-driven electricity shocks in IAMs or macroeconomic models, such as CGE models. Our work is in the spirit of Davis & Gertler (2015), who estimated the relationship between income, climate, and air conditioning for Mexico in 2010, and derives the implications in terms of aggregate electricity demand (Davis & Gertler, 2015). We are extending their analysis to three additional countries, we use more up-to-date data, and we use data over time so that we can take into account time-invariant, country-specific

unobservable characteristics and macro trends, such as technology evolution, and changes in aggregate market conditions and prices (see reviewer's comment 3).

Our work precisely goes in the direction of delivering new results that can be used to improve the characterization of the behavioral components of IAMs [Grubler et al. 2018]. We focus on a very specific aspect that is still missing in most IAMs and in all energy scenarios generated by models for the IPCC AR5 and the IPCC SR1.5: the impacts of climate change on energy demand in relation to the discomfort create by high temperature levels and specifically on AC demand. We agree that "IAMs are potentially powerful tools, and results from them have already been incorporated into IPCC Assessment Reports" [Grubler et al. 2018, but not including the impacts of climate change on energy demand can actually lead to a bias characterization of the mitigation challenges [Perera et al. 2020]. It is actually surprising that, despite the extraordinary growth the demand for cooling is experiencing, it is still a blind spot in the energy transition debate.

Here we propose two sets of results that can be used to improve space cooling and energy use for adaptation in IAMs: 1) country-specific AC adoption model and 2) climate-induced impacts on electricity demand. In the same spirit of van Ruijven et al. (2019), model (1) and shocks (2) are inputs that can be included in IAMs or Computable General Equilibrium (CGE) models [mostly shocks 2] or even in bottom-up models [model 1] to derive actual energy consumption that takes into account technology changes and price adjustments across multiple markets. For this reason, our models also avoid considering any direct or indirect adjustment induced by changes in technology and market conditions (prices), because these future changes in structural, technological and market characteristics can be better analyzed using IAMs or CGEs.

In order to clarify this twofold objective of this paper, we have included these considerations in the discussion of the revised manuscript (lines 321-327).

2) The second major concern is validation. Although validation of such integrated assessment procedures that link energy, future climate and cities are difficult [3] it is important that the authors provide sufficient confidence to the readers that the results obtained do make sense. For example, some of the assumptions made by the authors relating to cooling and energy demand for cooling are often used at top down models but lead to significant errors when you compute them using bottom up models. More importantly, such HVAC (heating ventilation and air-conditioning) demands are strongly influenced by the building physics which has not been considered at all. There are IAMs that use top down approaches. However, a clear validation about the model (if not all the elements but at least certain important parts within the model) is essential.

REPLY: We have added as section "Validation" in the Material and Methods section to clarify our validation strategy. The most important component of our model is the AC adoption model, which is based on a logistic regression that studies determinants of a dichotomic outcome (0,1), such as having or not having AC. When estimating models with a logistic function, the common practice of validation is to compute the Area Under the Receiver Operating Characteristics (ROC) Curve (AUC). Validation techniques for approaches based on logistic regressions exploit a classifier algorithm. Predicted probabilities are computed for all observations, and then the classifier algorithm assigns each predicted probability to class 0 or 1 based on a threshold (usually 0.5). If the predicted probability is larger than 0.5 the observation is classified in class 1, namely as having AC. If the predicted probability is smaller than 0.5 the observation is classified in class 0, namely as not having AC. The results are predicted classes for all the observations which are subsequently compared with the true observed classes in order to check the accuracy of the model. The goodness of a logistic regression is evaluated by building a confusion matrix, a table of fitted vs observed observation classes that allows to identify, after choosing the classification threshold, the number of false positive and negative the model predicts. Since the threshold choice for classification is arbitrary, the validation practice computes such confusion matrix for multiple thresholds and visualizes the results by using a ROC curve, a curve displaying the two types of errors for all possible thresholds. The overall performance of the logistic regression is evaluated over an

infinite number of thresholds by computing the area under the ROC curve, called AUC. The AUC has value between 0.5 and 1. The larger the AUC the better the performance of the logistic regression. As good practice, we first train our logistic regression on a training dataset defined as a random subsample of our dataset – containing 3/5 of total observations, and then we predict households with AC in the test dataset, the remaining subsample of 2/5 of total observations. For three out of four country-specific logistic regressions the AUC is more than 0.9, which is close to the maximum of one, so would be considered very good. It is 0.83 for Brazil which is still very good. Based on these results, we are quite confident in our projections, as our models well predict a household which owns an air-conditioning system.

3) Quantification of uncertainties Climate and energy related solutions always come up with uncertainties. In this case, future predictions for climate variation, technology evolution, market condition can come up with many uncertainties where uncertainty propagation plays a major role [4]. However, the model does discuss these uncertainties comprehensively.

REPLY: We do account for some drivers of uncertainty by using ensembles of climate and socio-economic models and scenarios, and our results are always presented in terms of ranges. We also address model uncertainty, as our estimates are presented with the standard errors and confidence intervals.

Still, we agree with the reviewer that addressing the uncertainties arising from the evolution of technology and market conditions is more difficult. We have added in the concluding section (from line 309) a more comprehensive discussion of those uncertainties, of how IAMs are a more suitable approach to address some of those, and of how our results can be used as inputs by models to develop these types of assessments.

References

- [1] Moss RH, Edmonds JA, Hibbard KA, Manning MR, Rose SK, Vuuren DP van, et al. The next generation of scenarios for climate change research and assessment. *Nature* 2010;463:747–56. <https://doi.org/10.1038/nature08823>.
- [2] Grubler A, Wilson C, Bento N, Boza-Kiss B, Krey V, McCollum DL, et al. A low energy demand scenario for meeting the 1.5 °C target and sustainable development goals without negative emission technologies. *Nature Energy* 2018;3:515–27. <https://doi.org/10.1038/s41560-018-0172-6>.
- [3] IAM helpful or not? *Nature Climate Change* 2015;5:81–81. <https://doi.org/10.1038/nclimate2526>.
- [4] Perera ATD, Nik VM, Chen D, Scartezini J-L, Hong T. Quantifying the impacts of climate change and extreme climate events on energy systems. *Nature Energy* 2020;5:150–9. <https://doi.org/10.1038/s41560-020-0558-0>.

Additional references used in the reply to the reviewer.

- Akpınar-ferrand, E., & Singh, A. (2010). Modeling increased demand of energy for air conditioners and consequent CO₂ emissions to minimize health risks due to climate change in India. *Environmental Science and Policy*, 13(8), 702–712. <https://doi.org/10.1016/j.envsci.2010.09.009>
- Auffhammer, M., & Mansur, E. T. (2014). Measuring climatic impacts on energy consumption: A review of the empirical literature. *Energy Economics*, 46, 522–530. <https://doi.org/10.1016/j.eneco.2014.04.017>
- Davis, L. W., & Gertler, P. J. (2015). Contribution of air conditioning adoption to future energy use under global warming. <https://doi.org/10.1073/pnas.1423558112>
- Isaac, M., & van Vuuren, D. P. (2009). Modeling global residential sector energy demand for heating and air

conditioning in the context of climate change. *Energy Policy*, 37(2), 507–521.

<https://doi.org/10.1016/j.enpol.2008.09.051>

McNeil, M. A., & Letschert, V. E. (2010). Modeling diffusion of electrical appliances in the residential sector. *Energy and Buildings*, 42(6), 783–790. <https://doi.org/10.1016/j.enbuild.2009.11.015>

Sailor, D. J., & Pavlova, A. A. (2003). Air conditioning market saturation and long-term response of residential cooling energy demand to climate change, 28, 941–951. [https://doi.org/10.1016/S0360-5442\(03\)00033-1](https://doi.org/10.1016/S0360-5442(03)00033-1)

van Ruijven, B. J., De Cian, E., & Sue Wing, I. (2019). Amplification of future energy demand growth due to climate change. *Nature Communications*, (Cmcc), 1–12. [https://doi.org/10.1038/s41467-019-10399-](https://doi.org/10.1038/s41467-019-10399-3)

Reviewer #2 (Remarks to the Author):

This study presents an empirical investigation on household’s adoption of air-conditioning in relation to climate and socioeconomics in four developing countries, and future scenarios. The paper advances the current state of the art by providing important insights on distributional aspects and inter- and intra-country differences in air-conditioning adoption and future space cooling deficit in developing countries. The manuscript is of interest for a broader audience and policy-relevant, clear and well written. My main comments are about comparison with a similar study, some methodological concerns, and strengthening of the conclusions.

Major comments:

- The authors should acknowledge that a similar empirical study exists for one of the investigated countries, Mexico (see reference below). It is recommended to revise the introduction and conclusions to highlight the advancements compared to this study and, if possible, add results comparison. Reference: Lucas W. Davis, Paul J. Gertler. Air conditioning and global warming. Proceedings of the National Academy of Sciences May 2015, 112 (19) 5962-5967; DOI: 10.1073/pnas.1423558112

REPLY: The reviewer is totally right and we actually apologize we eventually did not to include the Davis and Gertler paper that has been a key reference for our analysis. It was there since the early development of the work and then, somehow, was left uncited. We have added this paper, as well as some recent important references that have been published recently (e.g. Koshla et al 2021, ERL, Vigué et al ERL). We have also added a paragraph of comparison in the conclusions (line 292).

- The choice of using higher baseline temperatures for CDD calculation and use of wet-bulb versus dry-bulb CDDs are reasonable, in agreement with the indications in existing literature. However, I could not find in the text any justifications for the choice of 24°C as baseline temperature for both dry-bulb and wet-bulb CDDs. Using different baselines might lead to different results, depending on how temperature and humidity levels trigger air-conditioning adoption. I would therefore suggest to test how different baseline temperatures affect the model results and check to which extent results are robust across temperature levels and different countries. This could provide important additional insights and support the baseline temperature selection.

REPLY: We thank the reviewer for this comment. The use of wet-bulb (WB) as opposed to dry-bulb (DB) temperature better reflects the humid conditions of some of our countries. In order to better discuss how results vary across temperature measurement and threshold (which also answer the comment below), we have followed the reviewer’s suggestions and have compared 4 different metrics, adding two additional temperature thresholds compared to the original submission. We now perform a sensitivity analysis to a threshold of 22°C for both dry and wet-bulb. These temperature thresholds, combined with the dry- and wet-bulb measurements, give us a range of temperature thresholds between 22 and about 32°C DB, depending on humidity conditions (RH), as illustrated in the table below.

	22 Dry-Bulb (DB)
	24 DB
22 Wet-Bulb (WB)	25-30 DB (50% RH as in BRA and MEX-70% RH as in IND and IDN)
24 Wet-Bulb (WB)	28-32 DB (50% RH as in BRA and MEX-70% RH as in IND and IDN)

The sensitivity analysis suggests that, even within warm and tropical regions, temperature measurements based on DB can over-estimate the CDD elasticities in a way that depends on how AC is distributed across sub-regions with different micro-climates and humidity levels. Estimated CDD elasticities based on WB are smaller than CDD elasticities based on DB in all countries but Indonesia, where differences are actually small. The largest differences are found for Mexico and India. We believe these two countries show the largest discrepancy because they have a concentration of AC (higher AC adoption rates) in the regions characterized by a more arid climate (warm arid and very hot dry climate conditions). With respect to the sensitivity to the temperature thresholds, differences are smaller.

	AC-Brazil	AC-Mexico	AC-India	AC-Indonesia
22 deg - db	0.0774***	0.0578***	0.0469***	0.00350***
	(0.00222)	(0.00511)	(0.00469)	(0.00032)
24 deg - db	0.0608***	0.0533***	0.0444***	0.00339***
	(0.00194)	(0.00556)	(0.00440)	(0.00034)
22 deg - wb	0.0696***	0.0336***	0.0158**	0.00408***
	(0.00177)	(0.00680)	(0.00636)	(0.00040)
24 deg - wb	0.0565***	0.0230***	0.0172***	0.00373***
	(0.00154)	(0.00406)	(0.00588)	(0.00042)

When elasticities are combined with future CDDs, projections can be lower when using wet-bulb CDDs (AC in Mexico) because of the lower estimated elasticities, but they can also be higher (Brazil) because only slightly smaller elasticities interact with a larger increase in wet-bulb CDDs relative to the historical period. Since historical wet-bulb CDDs are much lower than dry-bulb CDDs, their growth rate is higher. Projections based on the 22°C temperature threshold tend to underestimate projections based on the 24°C temperature, especially when using WB measurements. We have included these results in the Supplementary Table 12 and in the Supplementary Figure 8.

Sensitivity of AC (0.50=50% share of households with AC). Boxplots show variations across SSPs and RCPs.

Sensitivity of electricity growth rate (1=100%). Boxplots show variations across SSPs and RCPs.

- Results based on both dry-bulb and wet-bulb CDD are provided, however there is no discussion on differences in outcomes between the two and whether accounting for humidity provides improved estimates in AC adoption in different countries.

REPLY: We have now added this sensitivity analysis in the paper. The additional analysis using the 22°C-threshold has helped us elaborate more robust considerations regarding the role of WB and DB. What we noticed is that dry-bulb temperature tends to overestimate the AC adoption elasticity to CDDs in particularly humid countries (India) or in countries characterized by high climate heterogeneities, such as Mexico. The new description is included at lines 173 and then in the conclusion starting at line 283.

- L.77. “We show that the growing penetration of air-conditioning will undoubtedly cause an upward pressure on aggregate electricity use”. This claim seems not entirely supported by the results, since the electricity model does not explicitly include air-conditioning adoption and it is not clear to what extent the increase in electricity demand might be driven by air-conditioning versus other appliances adoption.

REPLY: We thank the reviewer for this comment and we actually agree. We now present the results on AC and electricity as separate. We have rephrased the sentence as follows: “[...] We then evaluate how future changes in climate and socio-economic conditions will influence air-conditioning adoption, assuming people continue to adapt to climate conditions as they did in the recent past and as described by our empirical evidence. Using a top-down approach that extrapolates the historical evidence of long-run adaptation to the future, we also analyze how households will adjust electricity demand to changes in climate and income conditions.” Lines 54-58.

- L.278. “Growth factors vary significantly, from around threefold (across SSPs and RCPs) in Brazil...”. However, from Figure A.8 it seems that SSPs have a much greater influence on electricity changes than RCPs. The authors are invited to further elaborate on this (also related to my previous point) and report results for RCP4.5 in addition to RCP8.4 in Tables A.13-A.16.

REPLY: The reviewer is correct in the sense that SSPs have a greater influence on AC and ELY *changes* than RCPs. This partly reflects the relatively higher elasticities to income than CDDs. In order to better illustrate this point, we have conducted the ANOVA analysis.

When comparing the projections across RCPs, differences in the mean values of the distributions is only slightly statistically different (at 5%) for projected air-conditioning and not statistically significant for electricity growth rates.

When comparing the projections across SSPs, we find that the mean values of the distributions are statistically different (at a 1% level of statistical significance) for projected air-conditioning. Differences are driven by differences between SSP3- SSP 1, SSP 4- SSP 1, SSP 5- SSP 3, SSP 5- SSP 4.

When comparing the projections across countries and states by the five SSP5 groups, we find that the mean values of the distribution of electricity growth rates are statistically different. Differences are driven by differences between SSP 5- SSP 3, SSP 5- SSP 4.

We have added these considerations in the paper – line 297-300 and lines 260-261.

We have added RCP 4.5 in the Supplementary Tables now 14-21.

- A paragraph comprehensively describing the main limitations of this study and future research directions is missing in the discussion/conclusions section, and its addition is suggested.

REPLY: At lines 309-320 in the conclusions we now discuss the main limitation of our study.

Minor comments:

- L. 101: “households rarely use air-conditioning units”. Would “own” be more appropriate?

REPLY: Thank you, this sentence has been rephrased as suggested.

- L.124: “We estimate adoption models for air conditioners for each individual country by using the two most recent 124 survey waves available with a logit model”. Please, justify and provide more details in the

methods section on how multiple survey data were used in the estimation of the air-conditioning adoption model.

REPLY: The database assembled three waves for each country, with the exception of India. India was expected to release a new wave of the NSSO survey in 2020, but that did not happen. Being a comparative analysis, we have decided to use the same number of waves for all countries, therefore two most recent waves. We opted for using the two most recent waves because in these countries ownership rates are rapidly changing, and therefore they provide a more updated characterization of the current situation.

Depending on the wave used, the estimates capture different points on the logistic curve, being these countries on the convex part of the curve. Indeed, both CDD and income elasticities are larger when estimated using only the latest wave. This means that in the latest wave, *ceteris paribus*, households have a higher probability to adopt AC or, in other words, it is easier to adapt to climate change. We discuss this in the text (line 156 and Supplementary Table 11). Moreover, in the section on Methods we have added a section on data to clarify how different waves are used.

- Table 1: please, define in the caption the acronyms AC, FAN, and REF.

REPLY: Corrected as suggested

- Fig.5. It is suggested to specify in the caption that electricity use is total (not only air-conditioning).

REPLY: Corrected as suggested

Reviewer #3 (Remarks to the Author):

Overall, I found the new dataset collected and analyzed here to be a novel contribution, but a better elucidation of some of the strategies of the analysis and conclusions drawn would constitute an improvement in the paper.

The major claim of the paper is that it ‘brings new evidence on the specificity of air-conditioning dynamics and electricity use in Brazil, India, Indonesia and Mexico’. It also focuses on using this analysis to project future uptake of air conditioning, as well as the ‘adaptation cooling deficit’ presented by persistent low incomes and exacerbated by climate change. The collection and combination of extensive survey data for each country studied seems to be a new and important contribution to the field. The conclusions state that ‘both climate and income have comparatively similar roles in the increased adoption of air-conditioning in three of these countries. By contrast, findings from more developed countries suggest climate conditions play a relatively larger role than income...as well as the role of other variables.’ This type of conclusion strikes me as not particularly novel and indicates a more general aspect that it is not totally clear what important insights are made available by the large dataset and sophisticated statistical analysis presented by the paper.

For instance, Table 1 is a highly simplified distillation of adoption dynamics into only two variables (climate and income), and appears to tell most of the story. What’s less clearly presented is the relative importance and value of the many other variables included in the regressions, many of which (e.g. education level, urbanization, access to electricity) are clearly highly correlated. More discussion of this, and possibly Annex might help clarify this. Given the inherent uncertainties in forecasting economic development and climate, in my opinion the goal of such an analysis is to minimize the number of variables and clearly explain the explanatory value of those chosen and their limitations.

REPLY: We thank the reviewer for these two comments.

Regarding the innovation of the paper, we have rephrased our contribution by highlighting that we characterize the diversity of adoption dynamics for space cooling devices in emerging economies in relation to a much richer set of other variables, and we also characterize its distribution across sub-national states and income levels.

We have included these statements in the paper.

“we show that, in Brazil, India, Indonesia, and Mexico, income and humidity-adjusted temperature are common determinants of air-conditioning adoption, but their relative contribution varies in relation to households’ characteristics. Adoption rates are higher among households living in high-quality dwellings and in urban areas, and among those with higher levels of education.” [abstract]

“We show that in emerging economies the decision to purchase air-conditioning in response to warmer climatic conditions is strongly anchored to households’ socio-economic conditions, housing, and demographic characteristics. Variables indirectly related to wealth, such as housing conditions and education, play an important role across all countries, though the relative contribution of each factor is country specific.” [Introduction, line 70]

“While rising temperature and increasing income are likely to exert a positive pressure on the adoption and use of air-conditioning, here we show that the dynamics of air-conditioning are country-specific and relate to demographic and built environment characteristics, including education and housing conditions. Access to air-conditioning is highly uneven, indicating that households’ ability to adapt to climate change through the use of energy is linked to their socio-economic conditions. [conclusion, line 267]

Regarding Table 1, the idea of focusing on only those two variables was to put emphasis on the differences across countries and across appliance. We have included a clearer presentation of the relative importance and value of these variables to the many other variables included in the regressions. Moreover, we have modified Figure 2 to give more emphasis to an important result that was not much emphasized in the first submission, the result showing how the elasticity of adoption to CDDs varies with income.

We believe that it is important not to omit all the other covariates, which indeed correlated to the wealth conditions of households, because that would lead to a significant bias in both CDDs and total expenditure elasticities. We have added a table (Supplementary Table 11) with a comparison showing the bias, which is always positive and significant, especially for income. This means that projections based on elasticities estimated from a model that minimizes the number of variables would significantly overestimate the role of income, and also that of CDDs. A simpler model would omit variables that proxy for the wealth conditions of households (e.g. housing conditions and education) and urbanization, which also influences the actual local climate conditions, leading to a positive bias in the income and CDD elasticities (lines 155-156).

Following the reviewer's comments, we have also rephrased the aim of our projections as not implying to project future uptake of AC, but as to evaluate how changes in climate and socio-economic conditions with respect to the income will *influence* AC adoption, how many households will be left behind, and how electricity use will change, keeping everything else constant (lines 54-58). This assumption is not meant to suggest that other variables could not change or are less important, but it is meant to single out the contribution of these two specific drivers. The choice is motivated by 1) the inherent uncertainties in projecting economic development and wealth, for which we do not have a quantitative basis, 2) the usability of our results by those models that would like to include these mechanisms in model projections.

A more minor point, I was somewhat confused by the use of both fans and refrigerators included for comparison as 'cooling durables', since the service provided by these two are very different. Refrigerators are intuitively highly desirable in all climates, while fans are a space cooling 'proxy' for air conditioners and thus should have some degree of anti-correlation with them. Table 1 seems to indicate that the dependence on refrigerators on CDD is statistically significant, which is counterintuitive, and needs explanation.

REPLY: The reason for comparing AC adoption to that of both fans and refrigerators is that AC has characteristics that are similar to both types of goods. While fans are comparable to AC in terms of the space cooling service they provide, refrigerators are comparable in terms of type of good, requiring a larger expenditure/investment than fans. And indeed, when comparing the determinants, variables related to wealth are more important in the adoption decision of refrigerators than fans. For comparability, we have included the same set of variables across all goods, and indeed the marginal effect of CDDs on the adoption of refrigerators is very small, and its role is much smaller compared to expenditure, housing, and education. Although refrigerators are desirables in all climate, in poor regions where households might be used to consume fresh food long-term climate conditions can also affect the necessity of the good. Our results also show that, as income increases, refrigerators indeed become less sensitive to climate. They respond to CDDs at low income levels in Brazil and Mexico – where adoption is higher – and at medium income levels in India and Indonesia – where adoption is still quite low. We have added these considerations at lines 123-125 and lines 136-141, and Figure 2 now also shows how the average CDD elasticities of all three goods vary with income.

In the forecasting, a potentially significant variable in the forecasting of cooling deficit would be the real (PPP) cost of AC equipment, which has decreased rapidly in the past 20 years. This could be presented essentially as a sensitivity analysis.

REPLY: We agree, and indeed both AC and electricity prices could affect the adoption decision. Unfortunately, we do not have AC prices nor exogenous electricity prices that can inform our sensitivity analysis. Our estimates are only able to include fixed effects that are meant to capture the influence of states' fixed characteristics as well as time varying factors common to all states within each country. By comparing the results obtained with the latest wave (first columns in the table with Two Wave option "NO") with the results obtained with two waves, the table below indeed shows that the probability of adoption AC in the latest wave is higher, *ceteris paribus*. This means that, after controlling for all other variables including socio-economic conditions, adapting to climate change through the adoption of AC has become easier, for unobservable reasons. The unobservable conditions that have changed over time and that are not correlated with the rich set of covariates we include in the regression could indeed capture the lower cost of adaptation and therefore of AC. We have included now these results in the Supplementary Table 11 and at lines 157-162.

	AC-Brazil	AC-Brazil	AC-Mexico	AC-Mexico	AC-India	AC-India	AC-Indonesia	AC-Indonesia
mean_CDD_wb	0.000491***	0.000245***	0.000145***	0.000135***	0.000126***	0.0000931***	0.0000161***	0.0000114***
	(0.00002)	(0.00001)	(0.00003)	(0.00002)	(0.00004)	(0.00003)	(0.00000)	(0.00000)
ln_total_exp_usd_2011	0.186***	0.105***	0.0395***	0.0379***	0.0995***	0.0713***	0.0253***	0.0173***
	(0.00364)	(0.00167)	(0.00348)	(0.00329)	(0.00496)	(0.00374)	(0.00094)	(0.00065)
Cov	YES	YES	YES	YES	YES	YES	YES	YES
State FE	YES	YES	YES	YES	YES	YES	YES	YES
Year FE	NO	YES	NO	YES	NO	YES	NO	YES
Two waves	NO	YES	NO	YES	NO	YES	NO	YES

Finally, the placement of Materials and Methods at the end of the paper is somewhat strange to me. It might be clearer to put an abbreviated version of this up front, and include abbreviated details of dataset, functional forms and statistical results in the appendix.

REPLY: We agree, but that is the format of the journal. However, we have provided more details in the paper as well.

REVIEWERS' COMMENTS

Reviewer #1 (Remarks to the Author):

Acknowledge the reviewers for the comprehensive explanation.

Reviewer #2 (Remarks to the Author):

The authors carefully addressed all reviewers' comments, leading to substantial improvements in the manuscript. I have two remaining comments on the revised version:

- Electricity use. While the focus of the section "What drives the adoption of air-conditioning?" is on the AC adoption model results, results of the electricity quantity model (Table S9) are barely mentioned in the main text (L.151), before being used for the electricity use scenario results in the section "Adaptation cooling deficit". I would recommend to briefly describe the main findings from Table S9 beforehand, so that the drivers of electricity use are clear, and explain that electricity use is total and not only related to air-conditioning. I believe that this would help in supporting and clarifying the scenario results on electricity use.

- Dry/wet- bulb CDDs. The authors provided additional sensitivity results on the use of dry/wet bulb CDDs and different temperature thresholds, this is very much appreciated and brings some new initial insights on the role of humidity. I would suggest to briefly elaborate on the interaction effects between different CDD elasticities and future increase in dry/wet bulbs in the scenario results, as explained in the response to reviewer 2, and to consider whether to add such insights in the results section, rather than just in the conclusions.

Minor comments:

Line 31: The term "good dwellings" seems too generic. Perhaps replace with "higher quality dwellings"?

Line 55 "will influence air-the adoption of conditioning". Please, revise.

Line 142: "at middle- and low-income levels climatic conditions can make this good [refrigerators] more necessary." Not entirely clear. It is suggested to rephrase based on the response given to reviewer 3, e.g. refrigerators becoming less sensitive to climate at higher income levels.

Reviewer #3 (Remarks to the Author):

Changes made and detailed responses to my review questions largely address the concerns I had.
Recommend to publish without significant further changes.

Response to Reviewers:

Reviewer#2 comments

Electricity use. While the focus of the section “What drives the adoption of air-conditioning?” is on the AC

adoption model results, results of the electricity quantity model (Table S9) are barely mentioned in the main text (L.151), before being used for the electricity use scenario results in the section “Adaptation cooling deficit”. I would recommend to briefly describe the main findings from Table S9 beforehand, so that

the drivers of electricity use are clear, and explain that electricity use is total and not only related to airconditioning. I believe that this would help in supporting and clarifying the scenario results on electricity

use.

Response: Yes, we agree. We have added a description of the main findings starting at line 175 of the revised manuscript.

Dry/wet- bulb CDDs. The authors provided additional sensitivity results on the use of dry/wet bulb CDDs and different temperature thresholds, this is very much appreciated and brings some new initial insights on

the role of humidity. I would suggest to briefly elaborate on the interaction effects between different CDD

elasticities and future increase in dry/wet bulbs in the scenario results, as explained in the response to reviewer 2, and to consider whether to add such insights in the results section, rather than just in the conclusions.

Response: We have added this discussion starting at line 232, at the end of the “Future adoption of airconditioning around mid-century” section

Minor comments:

Line 31: The term “good dwellings” seems too generic. Perhaps replace with “higher quality dwellings”?

Response: Revised as suggested.

Line 55 “will influence air-the adoption of conditioning”. Please, revise.

Response: Revised as suggested.

Line 142: “at middle- and low-income levels climatic conditions can make this good [refrigerators] more necessary.” Not entirely clear. It is suggested to rephrase based on the response given to reviewer 3, e.g.

refrigerators becoming less sensitive to climate at higher income levels.

Response: Revised as suggested. Now line number is 147.